# Pneumatosis Intestinalis Induced by Alpha-Glucosidase Inhibitors in Patients with Diabetes Mellitus

**DOI:** 10.3390/jcm11195918

**Published:** 2022-10-07

**Authors:** Blake J. McKinley, Mariangela Santiago, Christi Pak, Nataly Nguyen, Qing Zhong

**Affiliations:** 1Department of Internal Medicine, Mayo Clinic, Jacksonville, FL 32224, USA; 2Department of Biomedical Science, Rocky Vista University College of Osteopathic Medicine, Ivins, UT 84738, USA

**Keywords:** pneumatosis intestinalis, diabetes, alpha-glucosidase inhibitors, acarbose, voglibose, miglitol, comorbidities, concomitant drugs, prednisone, immunosuppressants

## Abstract

Alpha-glucosidase inhibitor (αGIs)-induced pneumatosis intestinalis (PI) has been narrated in case reports but never systematically investigated. This study aimed to investigate the concurrency of PI and αGIs. A literature search was performed in PubMed, Google Scholar, WorldCat, and the Directory of Open-Access Journals (DOAJ) by using the keywords “pneumatosis intestinalis”, “alpha-glucosidase inhibitors”, and “diabetes”. In total, 29 cases of αGIs-induced PI in 28 articles were included. There were 11 men, 17 women, and one undefined sex, with a median age of 67. The most used αGI was voglibose (44.8%), followed by acarbose (41.4%) and miglitol (6.8%). Nine (31%) patients reported concomitant use of prednisone/prednisolone with or without immunosuppressants. The main symptoms were abdominal pain (54.5%) and distention (50%). The ascending colon (55.2%) and the ileum (34.5%) were the most affected. Nineteen (65.5%) patients had comorbidities. Patients with comorbidities had higher rates of air in body cavities, the portal vein, extraintestinal tissues, and the wall of the small intestine. Only one patient was found to have non-occlusive mesenteric ischemia. Twenty-five patients were treated with conservative therapy alone, and two patients received surgical intervention. All patients recovered. In conclusion, comorbidities, glucocorticoids, and immunosuppressants aggravate αGIs-induced PI. Conservative therapy is recommended when treating αGIs-induced PI.

## 1. Introduction 

Pneumatosis intestinalis (PI) is a condition in which gas is present within the walls of the intestines [1]. It is characterized by gas and free air in the mucosa, submucosa, and subserosa, and it can present in linear and/or cystic forms [2]. It is also called pneumatosis cystoides intestinalis. The incidence of PI is 2/6553 (0.03%) in autopsies [3]. There are two subtypes of PI: primary/idiopathic and secondary; secondary PI consists of up to 85% of all PI cases in adults [1,2].

PI is diagnosed via imaging techniques that include X-ray, computed tomography (CT), and endoscopy [4]. CT is the most sensitive medium for diagnosing PI.

PI can cause a wide range of symptoms with varying levels of severity. Some patients are asymptomatic, while others have life-threatening symptoms [5]. When patients with PI are asymptomatic, they may go undiagnosed [6]. On the other hand, the rupture of the subserosal cysts of PI could result in pneumoperitoneum without clinical peritonitis, and portal venous gas is often associated with pathological lesions [3]. Some PI cases may be secondary to transmural ischemia or necrosis of the gastrointestinal wall [7]. The mortality rates of PI increase in patients with bowel obstruction, toxic megacolon, cecal ileus, bone marrow transplants, and collagen vascular diseases [8].

Given that PI is a rare finding, its etiology is not well understood. However, PI has been reported in association with different disorders, including inflammatory bowel diseases, cytomegalovirus (CMV) colitis, acquired immunodeficiency syndrome (AIDS), emphysema, chronic obstructive pulmonary disease (COPD), cystic fibrosis, asthma, diabetes, cancer, organ transplants, fecal impaction, and mesenteric ischemia and necrosis [6,8,9].

While uncommon, some drugs are suspected of causing PI [8]. In particular, PI has been described in patients receiving corticosteroids, immunosuppressants, anticancer drugs, alpha-glucosidase inhibitors (αGIs), lactulose, and sorbitol [2,8,10]. In a multicenter study in Japan, out of 167 PI patients, 31 (19%) cases were related to diabetes. Among those 31 patients, 74.2% of them (23/31) had used αGIs [11].

αGIs competitively inhibit the intestinal α-glucosidase, thus delaying carbohydrate absorption in the small intestine. αGIs are commonly used to treat diabetes, especially type II diabetes. It has been thought that αGIs increase gastrointestinal luminal gas, contributing to the development of PI [12]. This raises a question: why does PI occur in some patients but not in others? So far, there have not yet been any systematic investigations on αGIs and PI. This study investigated the concurrency of PI and αGIs in patients with diabetes and sought to identify what other factors precipitate αGIs-induced PI.

## 2. Methods

### 2.1. Literature Search

A literature search was performed up to 7 June 2022, in PubMed/MEDLINE, WorldCat, Google Scholar, and the Directory of Open-Access Journals (DOAJ). We searched for articles using the following keywords: “pneumatosis intestinalis” AND “diabetes”, or “pneumatosis intestinalis” AND “alpha-glucosidase inhibitors”, or “acarbose” OR “voglibose” OR “miglitol” AND “pneumatosis intestinalis”. Inclusion criteria were the following: clinical trials/observational studies/case series or case reports that identified patients with intramural intestinal air, the usage of alpha-glucosidase inhibitors, and studies written in English. Exclusion criteria were the following: congress abstracts, and no alpha-glucosidase inhibitors involved.

All authors participated in the initial screening. Data extraction was performed by all authors and confirmed independently by two authors (B.M. and Q.Z.) to assess for correctness and bias. Data syntheses and analyses were performed by one author (Q.Z.) and confirmed by another (M.S.). Interpretations of the results were agreed upon by all authors.

### 2.2. Statistical Analysis

The characteristics of αGIs-induced PI cases were compared in patients with or without comorbidities, and between patients treated with voglibose and patients treated with acarbose, using the Student’s t-test or X^2^ test with a *p* < 0.05 as statistically significant.

## 3. Results

### 3.1. Articles Included

Our search resulted in 151 unique titles. All titles were screened, after which 30 full-text articles were assessed for eligibility. Ultimately, 28 articles with a total of 29 cases met our inclusion criteria, as shown in Figure 1. No clinical trials and observational studies were found. Details regarding patients’ information and characters are provided in Supplemental Appendix A [4,12,13,14,15,16,17,18,19,20,21,22,23,24,25,26,27,28,29,30,31,32,33,34,35,36,37,38].

### 3.2. Clinical Characteristics of PI

A total of 29 patients’ general information, comorbidities, past medical histories, and medication uses (Section 3.2, Section 3.3 and Section 3.4) are shown in Table 1. Twenty-three patients were diagnosed with type II diabetes, and six patients were diagnosed with steroid-induced diabetes as their diabetes occurred after prednisone use [23,25,26,28,34].

Eleven patients were men, 17 were women, and the sex of one patient was not defined. Ages ranged from 48 to 87, with a median of 67. The duration of time with diabetes varied from 2 days to 20 years.

Nineteen (65.5%) patients had comorbidities. Seven cases had connective tissue disorders/autoimmune diseases, including dermatomyositis [25], neuropsychiatric systemic lupus erythematosus [28], polymyalgia rheumatica [32], rheumatoid arthritis [34], granulomatosis with polyangiitis [34], hypothyroidism [22], and myasthenia gravis [26]. Three cases had immunocompromising conditions, including post-lung transplantation [38], minimal change disease-nephrotic syndrome [24], and non-specific interstitial pneumonitis (NSIP) [23]. Three patients had concomitant infections: one with acute cholecystitis [31], one with *E.coli* sepsis [24], and another with *Pseudomonas putida* detected in peritoneal dialysate effluent [37]. Five patients had hypertension, three of them with ischemic disease (post-cerebral infarction, vascular ischemia, heart ischemia, and/or nonocclusive mesenteric ischemia (NOMI)) and/or kidney failure [27,29,30,33,37].

Only 10 (34.5%) patients had no comorbidities. The number of patients with comorbidities was 1.9 times the number of patients without comorbidities.

### 3.3. Type of αGIs

All three available αGIs—acarbose, voglibose, and miglitol—were related to PI. The most common one was voglibose (44.8%), followed by acarbose (41.4%). There were only two cases of miglitol (6.9%). Two cases did not define a specific alpha-glucosidase inhibitor [17,30].

### 3.4. Concomitant Use of Other Drugs

As shown in Table 1, nine (31%) patients used prednisone or prednisolone ± other immunosuppressants or cytotoxic drugs [23,24,25,26,28,32,34,38]. Among those cases, one was combined with mizoribine (inhibiting guanosine synthesis), one with methotrexate, and one with tacrolimus [24,25,38]. Other antidiabetic drugs that were used consisted of insulin (seven cases), sulfonylurea (four cases), dipeptidyl peptidase-4 inhibitors (two cases), and metformin (one case). One patient used the carbohydrate supplement maltitol [22].

### 3.5. Symptoms

Characteristics of PI, including symptoms; diagnostic imaging; complications; the segments involved; the presence of free gas in cavities or other tissues; treatment; and outcomes (Section 3.5, Section 3.6, Section 3.7, Section 3.8, Section 3.9, Section 3.10 and Section 3.11) are shown in Table 2.

Patients presented with symptoms in 22/29 (75.9%) of the cases. As shown in Figure 2, the most common symptoms were abdominal pain (54.5%) and abdominal distention (50%), followed by diarrhea (22.7%), bloody stool (22.7%), and constipation (13.6%).

### 3.6. Segment of Bowel Involved

The location of PI had a wide distribution throughout the intestines. Nineteen (65.5%) cases had only large intestine involvement, seven (24.1%) had only small intestine involvement, and three (10.3%) had combinations of small and large intestine involvement. The ascending colon was the most involved (16/29, 55.2%), followed by the ileum (10/29, 34.5%).

### 3.7. Free Air in the Portal Vein and Intraabdominal Cavities and Extraintestinal Involvement

As shown in Table 3, patients with comorbidities had significantly higher rates of free air in the peritoneum or retroperitoneum (9/19 (47.4%) vs. 1/10 (10%), *p* < 0.05), and with small intestine involvement (8/19 (42.1%) vs. 2/10 (20%), *p* < 0.05), compared to patients without comorbidities. Only in patients with comorbidities was free air found in the mediastinum, pericardium, or subcutaneous space (2/19 (10.5%) vs. 0/10 (0%)) or the portal vein (3/19 (15.8%) vs. 0/10 (0%)).

### 3.8. Comparison between Patients Treated with Acarbose and Patients Treated with Voglibose

As shown in Table 1, the duration range for voglibose usage was 2 days to 10 years, with 6/13 (46.2%) having a duration shorter than 2 months. The duration range for acarbose usage was 1 year to 12 years, with 5/12 (41.7%) having a duration of less than five years. The median time of voglibose usage was relatively shorter than that of acarbose usage (0.6 years vs. 5 years).

As shown in Table 4, 7/12 (58.3%) patients who used acarbose and 10/13 (76.9%) patients who used voglibose had comorbidities. The durations of voglibose usage were shorter than those of acarbose regardless of the presence of comorbidities (without comorbidities: median 0.17 years vs. 3 years; with comorbidities: median 1.7 years vs. 8 years).

The voglibose group with comorbidities had a significantly higher ratio of concomitant usage of glucocorticoids compared to the acarbose group with comorbidities (80% vs. 14.3%, *p* < 0.01). The three cases of glucocorticoids + immunosuppressants were all in the voglibose group. Free air in the mediastinum, pericardium, or subcutaneous space only developed in the voglibose group with comorbidities (two cases, 20%).

### 3.9. Diagnosis

All cases of PI were confirmed by X-ray and computed tomography. Eleven (37.9%) patients were also checked with colonoscopy, and two patients were examined with endoscopic ultrasonography [26,35].

### 3.10. Treatment and Complications

The αGIs were recognized as potential causal drugs in all cases and were terminated. Twenty-five (86.2%) patients were only treated with conservative therapy: fasting was initiated at the onset of the disease in 12 (41.4%) patients, fluid supplementation in eight (27.5%) patients, antibiotic therapy in seven (24.1%) patients, and O_2_ therapy in six (20.6%) patients. One case delayed terminating the αGIs treatment until three months later, when the PI disappeared [35]. This same patient also received unique endoscopic needle puncture and high-frequency electro-scission of submucosal air cysts [35].

An exploratory laparotomy ruled out ischemia and necrosis in two (6.9%) patients with portal venous gas; therefore, no surgical intervention was needed, and the patients were treated with conservative therapy [31,33]. One case of portal venous gas was confirmed to have non-occlusive mesenteric ischemia by CT and endoscopic ultrasound examination, and the patient developed peritonitis and hypotension [37]. This case was treated with antibiotics, vasopressors, and continuous hemodiafiltration [37]. One patient developed *E. coli* sepsis that resulted in disseminated intravascular coagulation (DIC), acute renal failure, and acute respiratory distress syndrome [24]. This patient was treated with antibiotics, continuous hemofiltration, and mechanical ventilation [24].

Surgical intervention was performed in two cases. Laparotomy and hemicolectomy were performed on one patient [17], and a sigmoidectomy was performed on another to release a sigmoid volvulus [36].

### 3.11. Outcome or Duration of PI

All patients completely recovered. Symptom resolution ranged from 4 to 90 days, with a median of 7 days recorded in 16 (55.2%) patients. Resolution of PI was confirmed via images (CT and/or X-ray) or colonoscopy in 4 to 180 days, with a median of 18 days in 22 (75.9%) patients.

Although patients with comorbidities had increased rates of free air in cavities or extraintestinal tissues compared to patients without comorbidities, the median time of disappearance of PI was similar between those two groups, with a median of 21 to 21.5 days, as shown in Table 3.

## 4. Discussion

### 4.1. Mechanisms of αGIs-Induced PI

We summarized 29 cases of αGIs-induced PI. The number of patients with other comorbidities was 1.9 times the number of patients without comorbidities. The exact pathophysiological mechanism of PI is unclear, but there are three theories: “mechanical theory” (the air penetrates the bowel wall), “bacterial theory” (gas-forming bacteria penetrate the submucosa), and “biochemical theory” (luminal carbohydrates are fermented, increasing intraluminal pressure) [2,36]. Based on those theories, the authors propose that multiple factors contribute to the development of αGIs-induced PI: increased production of intestinal gas, the hypomotility of the gastrointestinal tract, weakened intestinal mucosa and wall, and/or air carried from the lungs. These mechanisms are depicted in Figure 3.

Increased production of intestinal gas could be due to αGIs, supplemental carbohydrates, and bacterial infection. αGIs lead to an increase in intestinal gas by suppressing carbohydrate absorption, resulting in the retention of carbohydrates in the lumen of the intestine. The unabsorbed carbohydrates are subsequently fermented by normal flora to produce carbon dioxide, methane, and hydrogen [22,35]. An additional carbohydrate supplement maltitol, a natural sweetener that is considered a sugar alcohol, or polyol may have played a concomitant role in one case of αGIs-induced PI as it is not readily absorbed in the small intestine and thus is fermented in the large intestine [22]. Three cases had bacterial infections [21,31,37]. Bacterial infection, especially gas-producing bacteria, not only increases gas production in the lumen of the intestine but also can invade and produce gas within the intestinal wall [39].

Hypomotility of the gastrointestinal (GI) tract will prolong the fermentation of carbohydrates and increase luminal pressure. In our series of cases, the hypomotility of the GI tract could be caused by diabetic autonomic nerve damage, hypothyroidism [22], dolichocolon (an abnormally long large intestine) [36], and atrophy and fibrosis of smooth-muscle cells of GI in dermatomyositis [25]. Increased intraluminal pressure could cause mechanical damage to the intestinal mucosa, resulting in gas migration into the intestinal wall [36].

The intestinal mucosa and wall could be weakened or damaged by inflammation, infection, or low blood perfusion. In our case series, there were two cases of colitis [4,35], three cases of infection [24,31,37], five cases of hypertension with or without vasculopathy (one of them with non-occlusive mesenteric ischemia), five cases of connective tissue disorders [25,28,32,34], and three cases of autoimmune diseases [22,23,26]. Bowel inflammatory diseases, including ulcerative colitis and Crohn’s disease, have been reported to be related to PI [9]. Hypertension and vasculopathy are important contributors to mesenteric ischemia in the elderly [16]. Connective tissue disorders could cause vasculitis or atrophy and fibrosis of smooth muscles of the GI tract, resulting in hypomotility [40].

Some drugs could also weaken or damage the intestinal mucosa and wall as well. In our series of cases, nine patients used prednisone/prednisolone, with three of them adding immunosuppressants or cytotoxic drugs [23,24,25,26,28,32,34,38]. Prednisone and immunosuppressants could deplete lymphocytes and shrink Peyer’s patches, facilitating gas entry into the intestinal wall [41]. Mizoribine and methotrexate are cytotoxic and may cause mucosa and epithelial cell damage directly [2]. In our case series, the patient with dermatomyositis treated with prednisolone and methotrexate was the only one developing subcutaneous air in the cervical area, along with pneumomediastinum, pneumoperitoneum, pneumoretroperitoneum, and PI [25].

Furthermore, a pulmonary cause needs consideration. Pulmonary diseases, such as chronic obstructive pulmonary disease, asthma, cystic fibrosis, and chronic cough, may force gas to enter the blood vessels and be carried from the lungs to the intestines [8]. In Hisamoto et al.’s study, the patient had non-specific interstitial pneumonitis; after using prednisone and voglibose, the patient presented with pneumomediastinum, pneumopericardium, pneumoretroperitoneum, and PI [23]. In Otsuka et al.’s study, the patient was a lung transplantation recipient for 1031 days and had pneumonia 47 days before finding PI [38]. In these two cases, chronic cough-induced alveolar rupture and PI could not be completely ruled out.

Patients with comorbidities not only have a higher risk of developing αGIs-induced PI but also have more severe imaging findings. Patients with comorbidities had a higher incidence of free air in cavities or extraintestinal tissues and a higher rate of small intestine involvement. Portal venous gas was only found in three patients, all of whom had comorbidities [31,33,37]. Only one of the three cases was confirmed to have non-occlusive mesenteric ischemia [37]. Although portal venous gas may suggest intestinal ischemia [42], it has been reported that 30% of patients with portal venous gas and PI were due to benign idiopathic causes [43].

### 4.2. Comparison of Three αGIs 

αGIs are especially effective in reducing postprandial hyperglycemia and are frequently used to treat patients with type II diabetes in combination with other antidiabetic drugs. The αGIs’ effect on lowering blood glucose is modest, but αGIs have the advantage of a very low risk of hypoglycemia compared to sulfonylurea drugs. αGIs often cause side effects of abdominal distention, flatulence, diarrhea, and abdominal pain. Because of the side effects, αGIs should not be used in patients who have gastrointestinal disorders [44,45,46].

PI is a rare side effect of αGIs. Voglibose and acarbose were more commonly reported than miglitol in causing PI. This may be due to different pharmacokinetics. Acarbose and voglibose are poorly absorbed in the intestine and primarily excreted in the feces, with approximately 30% that undergo fermentation by colonic microbiota [44]. In contrast, miglitol is absorbed by the gut and excreted unchanged in the kidneys [45]. Fermentation of voglibose and acarbose may increase luminal gas.

We found that PI developed more rapidly in patients treated with voglibose than in patients treated with acarbose. In our case series, patients with comorbidities on voglibose treatment had a higher ratio of simultaneous usage of glucocorticoids/immunosuppressants than those on acarbose treatment. As we discussed in Section 4.1, concomitant glucocorticoids/immunosuppressants may precipitate the development of PI. In addition, voglibose is 190 to 270 times more potent than acarbose [46]. Therefore, the authors hypothesize that a higher ratio of concomitant usage of glucocorticoids/immunosuppressants and a higher potency of voglibose contribute to a short duration of time that is needed to trigger PI compared to that with acarbose. Clinicians should be alerted that voglibose may cause PI in the first few months of its usage, whereas acarbose has the propensity to cause PI at any time during its usage, even a decade after its initiation.

### 4.3. Symptoms and Treatments

The most common symptoms are abdominal distension and pain, followed by diarrhea and bloody stool in the cases we analyzed. As these are non-specific symptoms, physicians should have a high level of suspicion for possible PI in patients that present with gastrointestinal complaints while taking αGIs. Additionally, physicians should be aware that PI may be identified incidentally on radiological imaging, endoscopy, or laparotomy in patients taking αGIs because several of the reviewed cases were asymptomatic.

The majority of patients fully recovered after conservative therapy, which includes the termination of alpha-glucosidases, fasting, fluid supplementation, antibiotics, and inhalation of oxygen. The authors recommend conservative treatment to be the mainstay of treatment in αGIs-induced PI.

We recommend sustaining from surgery, including exploratory laparotomies, if patients are stable without guarding or rebound tenderness. Taking into context the clinical presentation of the patients, PI and pneumoperitoneum or portal venous gas should not be used as an indication by themselves for an exploratory laparotomy. Additional criteria, including higher C-reactive protein concentrations, higher white blood cell counts, higher lactate levels, and ascites, may be required to indicate inflammatory syndrome and the likelihood of intestinal necrosis from mesenteric ischemia [7,47,48].

It is worth mentioning that although our case series had mild clinical features in most cases with patients recovering, severe complications from PI such as perforation and death have been reported in PI patients of other etiologies [2,7,47]. It is important to combine detailed history, laboratory, and image examinations to make differential diagnoses. While it is beneficial to avoid unnecessary laparotomy in patients with non-inflammatory signs, it is also important to keep patients whose clinical features worsen under careful observation.

### 4.4. Strength and Limitations

The strength of this study is its analysis of the contributions to PI from three alpha-glucosidase inhibitors, comorbidities, and other offending drugs. To our knowledge, this is the first thorough review of αGIs-induced PI.

This review, however, also has some limitations. Due to the low incidence rate, we cannot calculate the actual prevalence of αGIs-induced PI. Moreover, this study is only a description and observation, and no correlation or causative assessment has been generated. Furthermore, the possibilities of PI induced by other types of antidiabetic drugs have not been analyzed.

## 5. Conclusions

Alpha-glucosidase inhibitors are related to the development of PI. The most common ones are voglibose and acarbose. Voglibose usage may cause a much more rapid development of PI than acarbose. Patients with comorbidities and concurrent usage of glucocorticoids and/or immunosuppressants have a relatively higher risk of developing αGIs-induced PI with complications of free gas in cavities, portal veins, and extraintestinal tissues. The authors propose that multiple factors contribute to the development of PI when using αGIs. Intestinal ischemia or necrosis in αGIs-induced PI is uncommon, with only 1/29 (3.4%) patients having non-occlusive mesenteric ischemia in our study. The majority of patients recovered after conservative therapy. Therefore, the authors advocate for conservative therapy and the avoidance of any unnecessary surgery.

## Figures and Tables

**Figure 1 jcm-11-05918-f001:**
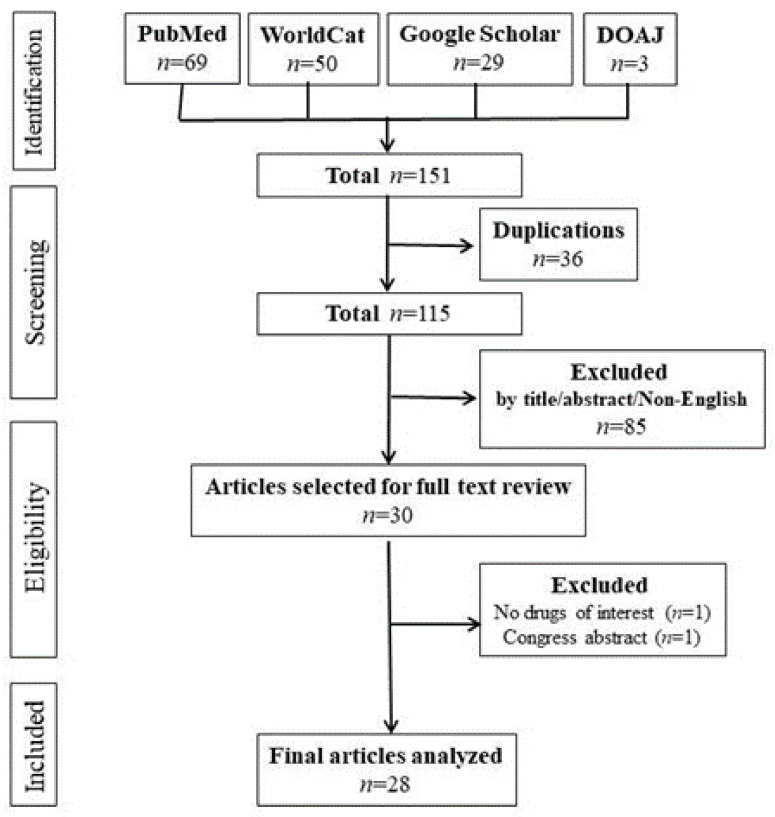
Literature search flow. Legend: A total of 151 abstracts were found from four databases of Pubmed, WorldCat, Google Scholar, and DOAJ. Twenty-eight articles met the inclusion criteria.

**Figure 2 jcm-11-05918-f002:**
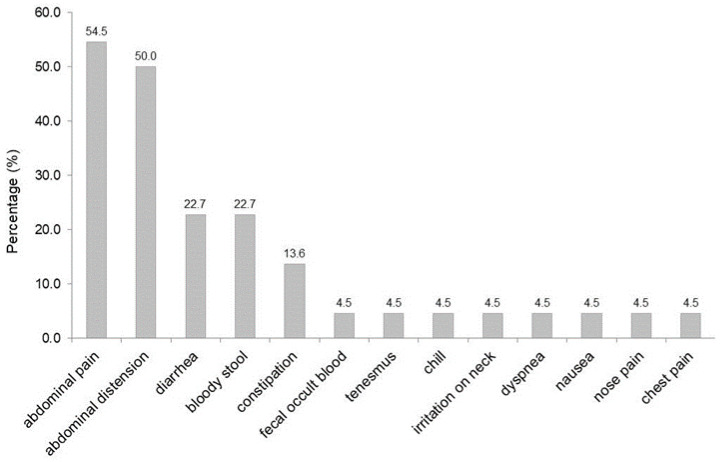
Overview of the symptoms of αGIs-induced PI. Legend: abdominal pain and distention were the most common symptoms, followed by diarrhea and bloody stool.

**Figure 3 jcm-11-05918-f003:**
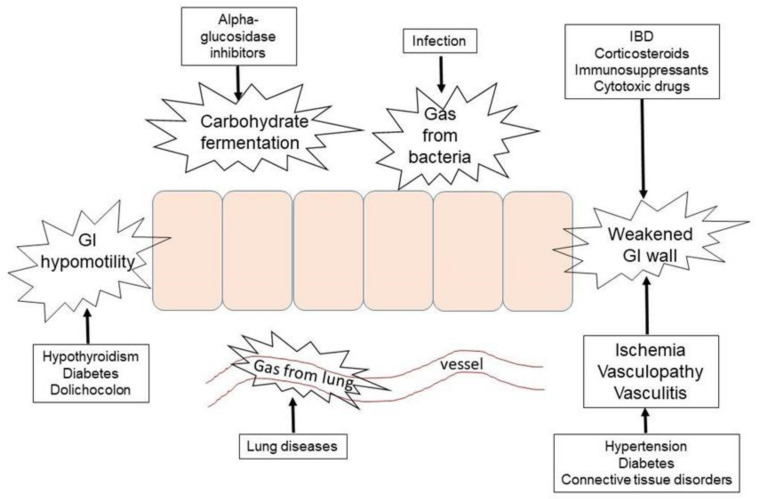
Possible mechanisms of αGIs-induced PI. Legend: There are multiple contributors to the development of αGIs-induced PI: increased production of intestinal gas, hypomotility of the gastrointestinal tract, weakened intestinal mucosa and wall, and/or air carried from the lungs. Abbreviations: IBD: inflammatory bowel disease; GI: the gastrointestinal tract.

**Table 1 jcm-11-05918-t001:** Clinical characteristics of patients with pneumatosis intestinalis.

Characteristics	*n*	(% of Cases)
Number of patients	29	
Men/women/undefined sex	11/17/1	
Age in years (mean ± SD) (median)	68.1 ± 10.3 (67)	
Age range	48–87	
Diabetes’ duration (mean ± SD) (median)	6.0 ± 6.1 (4)	
Range	2 days to 20 years	
Comorbidities and/or past medical history		
Number of patients	19	(65.5)
Connective tissue disorders/autoimmune diseases	7	(24.1)
Hypertension	2	(6.9)
Hypertension + post cerebral infarction	1	(3.4)
Hypertension + diabetic nephropathy + ischemic heart disease	1	(3.4)
Hypertension + diabetic nephropathy + peritonitis + nonocclusive mesenteric ischemia (NOMI) + ischemic disease + post cerebral infarction	1	(3.4)
Minimal change disease—nephrotic syndrome + *E. coli* sepsis	1	(3.4)
Chronic inflammatory colitis	2	(6.9)
Post lung transplantation + pneumonia 1 month prior	1	(3.4)
Sigmoid volvulus/dolichocolon	1	(3.4)
Non-specific interstitial pneumonitis (NSIP)	1	(3.4)
Acute cholecystitis	1	(3.4)
Medications		
Alpha-glucosidase inhibitors		
Acarbose	12	(41.4)
Median of duration (year) (Range)	5 (1–12)	
Voglibose	13	(44.8)
Median of duration (year) (Range)	0.6 (0.005–10)	
Miglitol	2	(6.9)
Median of duration (year) (Range)	3.8 (0.7–7)	
Undefined	2	(6.9)
Concomitant drugs/supplements		
Prednisone/prednisolone	6	(20.7)
Prednisolone + tacrolimus	1	(3.4)
Prednisolone + mizoribine	1	(3.4)
Prednisolone + methotrexate	1	(3.4)
Insulin	7	(24.1)
Sulfonylurea	4	(13.8)
Dipeptidyl peptidase-4 inhibitors	2	(6.9)
Metformin	1	(3.4)
Maltitol	1	(3.4)

**Table 2 jcm-11-05918-t002:** Characteristics of PI.

Characteristics	n	(%)
Symptoms		
Asymptomatic	7	(24.1)
Symptomatic	22	(75.9)
Imaging		
Abdominal X-ray	29	(100)
Abdominal CT	29	(100)
Colonoscopy	11	(37.9)
Segments involved		
Large bowel only		
Ascending colon only	5	(17.2)
Sigmoid only	5	(17.2)
Ascending + sigmoid	1	(3.4)
Ascending + transverse colon	2	(6.9)
Ascending + descending colon	2	(6.9)
Cecum + splenic flexure colon	1	(3.4)
Cecum + ascending + transverse + sigmoid colon	1	(3.4)
All colon	2	(6.9)
Small intestine only		
Ileum only	1	(3.4)
Whole small intestine	6	(20.7)
Combined		
Ileum + ascending colon	2	(6.9)
Ileum + ascending + transverse colon	1	(3.4)
Free gas in cavities or other tissue		
Pneumoperitoneum	7	(24.1)
Pneumoretroperitoneum	2	(6.9)
Portal venous gas	2	(6.9)
Portal venous gas + pneumoperitoneum	1	(3.4)
Subcutaneous air in the cervical region + pneumomediastinum + pneumoretroperitoneum + pneumoperitoneum	1	(3.4)
Pneumomediastinum + pneumopericardium + pneumoretroperitoneum	1	(3.4)
Treatment		
Termination of alpha-glucosidase inhibitors	29	(100)
Conservative	25	(86.2)
Fasting	12	(41.4)
Fluid supplementation	8	(27.6)
Antibiotics	7	(24.1)
Oxygen therapy		
Conventional	5	(17.2)
Mechanical	1	(3.4)
Endoscopy (colonoscopy) therapy		
Needle puncture + electro-resection of gas cysts	1	(3.4)
Hemofiltration	2	(6.9)
Exploratory laparotomy but with conservative therapy	2	(6.9)
Laparoscopic sigmoidectomy	1	(3.4)
Laparotomy and hemicolectomy	1	(3.4)
Outcome		
Survival	29	(100)
Free air disappearance was confirmed radiologically	22	(75.9)
Median of duration in days (range)	18 (2-180)	

Abbreviations: CT: computed tomography; PI: pneumatosis intestinalis.

**Table 3 jcm-11-05918-t003:** Comparison between patients with and without comorbidities.

	Patient without Comorbidities (*n* = 10)	Patients with Comorbidities (*n* = 19)
Age (years) (mean±SD) (median)	65.5 ± 8.4 (64.5)	69.6 ± 11.0 (70)
Pneumoperitoneum or pneumoretroperitoneum	1	9 *
Pneumomediastinum or pneumopericardium or subcutaneous air	0	2
Portal venous gas	0	3
Small intestine involvement	2	8 *
Combination of small and large intestines	1	2
Exploratory laparotomy	1	2
Surgery	1	1
PI disappearance (days) (median) (range)	21.5 (4–180)	21 (4–90)

* *p* < 0.05 compared to patients without comorbidities.

**Table 4 jcm-11-05918-t004:** Comparison between patients who used acarbose and patients who used voglibose.

	Acarbose		Voglibose	
	Without Comorbidities	With Comorbidities	Without Comorbidities	With Comorbidities
Number of patients	5	7	3	10
Age (years) (mean±SD) (median)	63.6 ± 8.2 (65)	72.6 ± 9.3 (72)	67.7 ± 9.1 (64)	64.8 ± 10.1 (69.5)
Diabetes’ duration (years) (mean ± SD) (median)	Unknown	9.8 ± 3.5 (10)	11.5 ± 12 (11.5)	1.1 ± 1.6 (0.08)
αGIs duration range (years) (median)	1–10 (3)	2–12 (8)	0.05–5 (0.17)	0.005–10 (1.7)
Concomitant prednisone/prednisolone ± immunosuppressants (case) (%)		1 (14.3%)		8 (80%) **
Portal venous gas		1 (14.3%)		2 (20%)
Pneumoperitoneum +/− Pneumoretroperitoneum		3 (42.9%)		5 (50%)
Pneumomediastinum, Pneumopericardium, Pneumoretroperitoneum				1(10%)
Subcutaneous air in the cervical region, pneumomediastinum, pneumoperitoneum, pneumoretroperitoneum				1(10%)
Exploratory laparotomy		1 (14.3%)		1(10%)
Laparoscopic sigmoidectomy		1 (14.3%)		

** *p* < 0.01 compared to the acarbose group with comorbidities.

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
