# Peer review of "Pneumatosis Intestinalis Induced by Alpha-Glucosidase Inhibitors in Patients with Diabetes Mellitus"

_jcm, 2022, doi:10.3390/jcm11195918_

Round 1

Reviewer 1 Report

In this descriptive study the authors investigated the concurrency of pneumatosis intestinalis (PI) and alpha-glucosidase inhibitor (αGIs) in diabetic patients and sought to identify what other factors precipitate aGIs-induced PI. In total, 29 cases of  αGIs-induced PI in 28 articles were included. Authors found that the most used αGI was voglibose (13 patients), followed by acarbose (12 patients) and miglitol (2 patients). The main symptoms were abdominal pain (54.5%) and distention (50%). The ascending colon (51.7%) and the ileum (34.5%) were the most affected. Nineteen (65.5%) patients had comorbidities. The majority of patients (25) recovered after conservative therapy.

In my opinion, the manuscript is interesting, is well prepared and written, but the authors should make some minor modifications.

-Table 1 indicates that the median of duration (year) (range) of acarbose was 5 (2-12) but in the text (lines 152-153) it is indicated that the duration range for acarbose usage was one year to 12.

-Line 160. "Other antidiabetic drugs included insulin (7 cases), sulfonylurea (2 cases)".  However, table 1 shows 4 cases of sulfonylureas (13.8%)

-In Table 2, the sum of patients according to segments involved is 28 patients, not 29: large bowel only (18), small intestine only (7) and combined (3). There are also 28 patients as reported on lines 202-204.

-I think it would be interesting to create a new table comparing the clinical characteristics and comorbidities between the patients taking voglibose (13 patients) and acarbose (12 patients). This comparison could rule out whether, in addition to the different pharmacokinetics of both drugs, there are other possible explanations for the earlier appearance of PI with the intake of voglibose

Reviewer 2 Report

Dear editor,

I have review the manuscript "Pneumatosis intestinalis induced by alpha-glucosidase inhibitors in diabetic patients”.

The aim of the present study was to investigate the 29 concurrency of PI and αGIs.

The results are well presented, discussed in detail.
Methods are appropriated.
The discussion section of the manuscript is well structured and comprehensive.

 Although it is a work that could potentially be of interest, it needs some changes.

 General comments.

Authors are kindly requested to change the term “diabetic patients” to “patients with diabetes.”

It would also be interesting to provide data on the use of alpha-glucosidase inhibitors and to provide some line of discussion and even limitations regarding the use of these drugs in clinical practice.
